# Role of Inflammatory Factors during Disease Pathogenesis and Stem Cell Transplantation in Myeloproliferative Neoplasms

**DOI:** 10.3390/cancers12082250

**Published:** 2020-08-12

**Authors:** Nicolas Chatain, Steffen Koschmieder, Edgar Jost

**Affiliations:** Department of Hematology, Oncology, Hemostaseology and Stem Cell Transplantation, Faculty of Medicine, RWTH Aachen University, 52074 Aachen, Germany; nchatain@ukaachen.de

**Keywords:** myeloproliferative neoplasm, inflammation, cytokines, allogeneic hematopoietic stem cell transplantation, JAK inhibitors

## Abstract

Hematopoiesis is a highly regulated and complex process involving hematopoietic stem cells (HSCs), cell surface adhesion molecules, and cytokines as well as cells of the hematopoietic niche in the bone marrow (BM). Myeloproliferative neoplasms (MPNs) are characterized by clonal expansion of HSCs involving one or more blood cell lineages. Philadelphia-negative MPNs (Ph-neg MPNs) comprise polycythemia vera (PV), essential thrombocythemia (ET), and primary myelofibrosis (PMF). In nearly all patients with Ph-neg MPN, mutations in the genes encoding janus kinase 2 (JAK2), calreticulin (CALR), or the thrombopoietin receptor (MPL) can be detected and, together with additional mutations in epigenetic modifier genes, these genetic aberrations contribute to the clonal expansion of the cells. In addition to these intracellular changes in the malignant clone, inflammatory processes involving both the clonal and the non-clonal cells contribute to the signs and symptoms of the patients, as well as to progression of the disease to myelofibrosis (MF) or acute leukemia, and to thrombotic complications. This contribution has been corroborated in preclinical studies including mouse models and patient-derived iPS cells, and in clinical trials, using anti-inflammatory drugs such as JAK inhibitors and steroids, or immunomodulatory drugs such as IMiDs and interferon-alpha (IFNa), all of which change the (im)balance of circulating inflammatory factors (e.g., TNFa, IL-1b, and TGFβ) in MPN. Currently, allogeneic hematopoietic (stem) cell transplantation (allo-HCT) remains the only curative treatment for Ph-neg MPN and is the treatment of choice in intermediate-2 and high-risk MF. HCT can reverse inflammatory changes induced by MPN as well as fibrosis in a large proportion of patients, but it also induces itself profound changes in inflammatory cells and cytokines in the patient, which may help to eradicate the disease but also in part cause significant morbidity (e.g., by graft-versus-host disease). In this review, we focus on the contribution of aberrant inflammation to disease pathogenesis in Ph-neg MPN as well as the current understanding of its alterations after allogeneic HCT.

## 1. Background

Myeloproliferative neoplasms (MPNs) are clonal diseases of the bone marrow (BM). Because the Philadelphia chromosome (Ph+) resulting in the BCR-ABL fusion protein presents a unique disease cause and therapeutic target, chronic myeloid leukemia (CML) is set apart as the only member of Ph+ MPN in the World Health Organization (WHO) classification of MPN. The other classical MPNs are Philadelphia-negative (Ph-neg) and comprise polycythemia vera (PV), essential thrombocythemia (ET), and primary myelofibrosis (PMF), and these Ph-neg MPNs are the topic of this review. While most Ph-neg MPN patients harbor mutually exclusive somatic mutations in the genes of *JAK2* (exon 12 or 14), *CALR* (calreticulin–CALR protein; frameshift mutation in exon 9), or the thrombopoietin receptor *MPL* (exon 10) [1], 10–15% of PMF and ET patients are termed to have triple-negative MPN because they do not show any of these so-called driver mutations [2]. In recent years, several other so-called atypical driver mutations have been identified in the genes of *JAK2* and *MPL* of triple-negative MPN patients [3,4,5]. Over 95% of PV patients and over 50% of both ET and PMF patients carry the JAK2V617F mutation, which is situated in the inhibitory pseudokinase domain of JAK2 and leads to cytokine-independent JAK/STAT signaling pathway activation. Similarly, point mutations in the juxtamembrane domain of MPL at position W515 lead to constitutive activation of receptor-associated JAKs (particularly JAK2), and mutated CALR leads to JAK/STAT activation through oligomerization and activation of MPL. In addition to the aforementioned aberrations, common so-called non-driver mutations were found in genes of epigenetic modifiers (i.e., *ASXL1*, *TET2*, and *EZH2*), factors involved in spliceosome function (i.e., *SRSF2*, *U2AF1*, and *SF3B1*) as well as metabolic modifiers and linker proteins (i.e., *IDH2*, *SH2B3*, and *CBL*) [6,7]. The prognosis of MPN patients strongly depends on the type and number of these additional mutations [6]. 

Biologically, MPN can be viewed as a state of chronic inflammation maintained by not only the malignant clone but also by the cells of the BM microenvironment, including mesenchymal stromal cells (MSCs), endothelial cells (ECs), and osteoblasts (OBCs) [8,9,10,11], and by the non-clonal hematopoietic cells (HCs), in a process of corruption by the malignant cells [12]. Cytokine receptors involved in inflammation are typically associated with a unique display of janus kinases (JAK1, JAK2, JAK3, and TYK2), which then become activated upon ligand binding and drive signal transduction from the cytoplasm to the nucleus, among others by activation of the signal transducer and activator of transcription family (STAT1, STAT2, STAT3, STAT4, STAT5A, STAT5B, and STAT6). Therefore, JAKs play crucial roles in acute and chronic inflammation [13,14,15]. The resulting activated sets of genes include many inflammatory factors, some of which are secreted into the extracellular space (e.g., CALR [16]) where they stimulate the local microenvironment. Accordingly, mutation and constitutive activation of JAK2 in MPN is prone to generate an inflammatory milieu inside and outside of the malignant clone [17,18,19]. Importantly, chronic inflammation is a risk factor for clonal evolution and may be a trigger of transformation into myelofibrosis (MF) and acute myeloid leukemia (AML) [20,21]. 

Hematopoietic (stem) cell transplantation (HCT) was pioneered in the 1970s and is now established for a wide variety of hematopoietic malignancies, including MF [22,23,24,25], where it is established as a standard treatment for intermediate-2 and high-risk PMF, post-PV MF, and post-ET MF. Most commonly, the risk stratification is currently calculated by the Dynamic International Prognostic Scoring System (DIPSS) [26,27] or DIPSS-plus score [28], and molecular characteristics of the disease [29]. HCT remains the first choice treatment for transplant-eligible patients with intermediate-2 or high-risk MF [30,31] and the only curative treatment for MF [32,33,34]. 

In this review, we summarize the prominent cell types and known soluble factors fueling inflammation in MPNs. Furthermore, we discuss the available data on the impact of HCT on inflammatory cytokines, cells, and fibrotic changes in the BM microenvironment.

## 2. Inflammatory Cytokines and Their Sources in MPN

A plethora of cytokines regulate hematopoiesis and differentiation of distinct cell populations, serving as a rampart against infections, preventing excessive blood loss by enhancing coagulation, and orchestrating the BM niche and vascularization (Figure 1). In MPNs, many cytokines are upregulated, creating a sustained inflammatory microenvironment [35]. In addition, most patients with MPNs, particularly MF, display splenomegaly and splenic extramedullary hematopoiesis, and radionuclide studies have confirmed that the spleen is an important source of inflammatory factors [36,37].

Tefferi et al. analyzed plasma of 127 PMF patients and confirmed an increase of 19 circulating cytokines in comparison to healthy controls [38]. Among them, interleukin-8 (IL-8), sIL-2R, IL-12, IL-15, and inducible protein 10 (IP-10) were independently predictive for inferior survival. For sIL2-R, IL-8, IL-12, and IL-15, the prognostic relevance for overall survival was maintained in multivariate models including DIPPS score. Leukemia-free survival was strongly associated with elevated IL-8 expression. In PV, increased macrophage inflammatory protein 1b (MIP1b) levels were correlated with shortened survival, and significant correlations were described for interferon-alpha (IFNa) and IFNg with thrombocytosis, IL-12 levels and hematocrit, as well as IL-1β, IL-2, IL-7, fibroblast growth factor b (FGFb), and hepatocyte growth factor (HGF) with leukocytosis [39].

TNFa, IL-6, and TGFβ plasma levels have been analyzed in further studies, and significant upregulation as well as signaling pathway activation were demonstrated for these factors [40,41,42]. Fleischman et al. demonstrated that the presence of TNFa was required to initiate an MPN phenotype in a murine retroviral BM transplantation model of JAK2V617F-induced MPN [42]. Mechanistically, TNFa stimulated colony growth of JAK2V617F MPN cells, while colony numbers of JAK2 wild-type (WT) cells were reduced, demonstrating how MPN clones may acquire a selective growth advantage over their normal counterparts [42,43]. Similar observations were made for lipocalin-2 (LCN2 or neutrophil gelatinase-associated lipocalin (NGAL)), which is upregulated in MPN and was shown to induce cell death of normal HCs but to spare or promote CD34+ MF cells [44,45]. At the same time, LCN2 generates an aberrant microenvironment that also suppresses normal hematopoiesis but enhances survival of the malignant clone [46], e.g., via stimulation of fibroblasts and other BM stromal cells to produce excess fibers. 

Dysplastic megakaryocytes, platelets and LepR+ or Gli1+ MSCs are critically involved in the manifestation of MF [47]. On the one hand, megakaryocytes and platelets stimulate the malignant MPN clone itself via secretion of inflammatory cytokines such as IL-6, TGFβ, PDGF, TNFa, and IFNa. On the other hand, via CXCL4, they also recruit and activate cells of the BM microenvironment such as LepR+ and/or Gli1+ MSCs, thus triggering myofibroblast formation and BM fibrosis [48,49]. Subsequently, downregulation of the key hematopoietic stem cell (HSC) maintenance factors CXCL12 and SCF and deposition of fibrinogen as well as collagen are promoted, leading to exhaustion of normal hematopoiesis and resulting in anemia, thrombocytopenia, and neutropenia. Figure 1A summarizes the inflammatory microenvironment in MPN in the absence of allogeneic (allo-) HCT.

In addition to circulating cytokines maintaining chronic inflammation, CD34+ HSCs as well as PBMCs of PV, ET, and PMF patients show transcriptomic deregulation of genes related to inflammation, immune response, and DNA repair [50,51,52,53]. While chronic inflammation and upregulation of the aforementioned pathways favor fibrosis, we and others have described an interferon-stimulated gene expression pattern, priming most effectively JAK2V617F-positive MPN cells for IFNa treatment in comparison to CALR-mutated cells [52,54] and highlighting important differences in inflammatory signaling and therapy response associated with the different driver oncogenes. 

Recently, Wong et al. published data on the analysis of differential gene expression profiles in patients with MF grade 0–III [55]. RNA was isolated from histologic sections, and four clusters were identified by hierarchical clustering and gene set enrichment analysis (GSEA), showing increased expression of inflammatory genes, especially in overt MF, with excessive IFN, TNFa, and mature dendritic cell (DC) gene signatures. Nevertheless, the inflammatory gene expression pattern differed from MF patient to patient, suggesting that the common phenotype of MF may arise from distinct inflammatory pathways [55]. Importantly, the study by Wong et al. supports the idea of an inflammatory microenvironment that initiates and alters the fibrotic progression.

The increase in DC gene signatures observed by Wong et al. may suggest a novel crucial role of DCs in MF progression. DCs are major antigen-presenting cells (APCs), orchestrate adaptive immune responses, and secrete inflammatory cytokines implicated in fibrosis, e.g., IL-1, IL-6, IL-12, IL-23, IFN, and TNFa [56]. Conversely, Romano et al. observed significantly lower amounts of circulating myeloid DCs in JAK2V617F-positive MF patients in comparison to healthy persons and impaired in vitro DC differentiation capacity of monocytes [57]. Furthermore, DCs deficient in antigen presentation (MHCII knockout driven by the CD11c promoter), and consequently the missing collaboration between DCs and CD4+ T cells, led to a myeloproliferative disorder in mice [58]. These data suggest that the effect of DCs on MPN/MF development relies more on the regulation of the immune response via CD4+ T cells than on the participation in the “cytokine storm”. 

However, not only cytokines fuel inflammation and support the malignant clone. Reactive oxygen species (ROS), such as H_2_O_2_ and O_2_^−^, have diverse functions in healthy BM, e.g., maintaining quiescence, self-renewal, and the differentiation potential of hematopoietic stem and progenitor cells (HSPCs), but were also described to be involved in disease progression of MPN (reviewed in [59]). Chronic oxidative stress may trigger DNA double-strand breaks and DNA damage and induce aging or carcinogenesis [59,60,61]. Nevertheless, the functions of ROS remain controversial, as relatively normal levels of ROS can stimulate the proliferation of aberrant cells, and an optimized detoxification of ROS is characteristic for the success of the malignant clone [62,63]. In JAK2V617F-positive MPN, ROS levels were shown to be elevated and data of increased events of double-strand breaks (DSBs) and DNA repair suggest the same for CALR-mutated MPN cells [64]. Using a murine JAK2V617F knock-in model, Marty et al. demonstrated the upregulation of ROS in the HSC compartment and resultant increase of DNA damage [65]. In addition, when JAK2V617F-KI mice were treated with the antioxidant N-acetylcysteine (NAC), splenomegaly, JAK2V617F-postive cell infiltration, and DNA damage were reduced in these mice. Similarly, ROS levels were shown to be high in JAK2V617F-positive cells in comparison to JAK2 WT cells, and increased ROS led to stabilization of the hypoxia-inducible factor 1a (HIF1a) protein, which was shown to be crucial for the survival of JAK2V617F-expressing cells [66]. Hence, targeting of ROS and/or HIF1a, in addition to JAK inhibition, could be beneficial in JAK2V617F-positive MPN. Just recently, Carver et al. demonstrated the potential of NAC to reduce the thrombotic risk in a JAK2V617F mouse model and the initiation of a clinical trial was considered [67].

## 3. Allogeneic HCT: Conditioning, Transplantation, and Post-HCT Management

HCT is a complex procedure, and several of its phases induce or affect inflammatory processes as follows: pretransplant phase, conditioning, stem cell infusion, homing, aplasia, regeneration, as well as graft-versus-leukemia (GVL) effects and remodeling of the BM (graft-versus-myelofibrosis) (Figure 2). During HCT, the malignant clone composed by HCs is replaced by normal hematopoietic stem and progenitor cells (HSPCs) as well as immature and mature immune effector cells of the donor. In addition to the myeloid compartment, these immune effector cells of the donor, mainly T lymphocytes, will also become active in the recipient. Peripheral blood-derived stem cells (PBSCs) are the main source of the graft. As a consequence, no significant number of BM stromal cells, such as MSC, adipocytes, ECs, or OBCs, will be transfused. Therefore, the course of a patient during and after HCT is a clinical model to study the interaction of the new healthy donor-derived HCs with the remaining stromal cells and the stem cell niche of the patient, as compared to previous influences by the malignant HC population and constant inflammatory stimuli [47], which typically had evolved for several years. Even in patients in whom the stromal cells do not harbor the driver mutation that is present in the malignant clone [68], genetic aberrations and epigenetic reprogramming of these cells have been described in MPNs and other myeloid neoplasms such as acute leukemias [69,70]. 

MF is associated with extramedullary hematopoiesis, progressive cytopenias, and accumulation of fibrotic tissue composed of reticulin/collagen fibers (Figure 2) [71]. Fibrosis of the BM was long considered irreversible, and, thus, there were concerns about higher treatment-related mortality due to graft failure related to fibrosis and splenomegaly in patients affected by MF. However, analysis in HCT patients has shown that BM fibrosis, which is closely connected to inflammatory cells and cytokines, as shown above, is a highly dynamic process that is typically reversible upon HCT and achieved through a T cell-mediated “graft-versus-myelofibrosis” effect, leading to a progressive reduction in marrow fibrosis from grade III to grade 0 (Figure 2) [72,73]. This process usually takes several months but has been observed as rapidly as 30 days after HCT [74]. While relapse of MF after HCT is often associated with an increase of JAK2V617F mutant allele burden, subsequent restoration of molecular remission (MR) has been observed after donor lymphocyte infusion (DLI) [75,76]. Importantly, the reversal of fibrosis and restoration of MR can be explained by the interaction of the normal HCs with the stromal cell types and extracellular matrix in the BM via cell–cell interaction as well as alterations of circulating cytokines and other inflammatory factors (summarized in Figure 2). In the following sections, some of these inflammatory factors are highlighted and the impact of new treatment modalities before and after HCT on the inflammatory microenvironment, pathophysiology, and outcome of patients are discussed, taking into account that the manifold interactions and effectors are not fully understood. Figure 2 provides an overview of the temporal progress and treatment options pre and post allo-HCT.

## 4. Inflammatory Conditions in the Hematopoietic Niche before and after HCT

Similar to age-associated skewing (AAS) of the X-chromosome in HCs, the entire BM niche is subject to changes in the elderly population. Reduction of the endosteal niche, which is associated with age-associated lymphoid-to-myeloid skewing [82], remodeling of osteoblast function (e.g., osteoporosis) [83], as well as the replacement of hematopoietic red BM by fatty non-hematopoietic yellow marrow [84,85], are some examples. In MPN, these alterations are pronounced, providing excellent growth conditions for the malignant clone and its progenies, and sheltering them from treatment-induced apoptosis. Schepers et al. demonstrated a switch from a normal HSC-supporting niche to a leukemic stem cell (LSC) “service station” using a BCR-ABL-expressing mouse model [86]. Osteoblastic lineage cells were functionally altered by the malignant cells to both reduce the interaction and support of normal HSCs and stimulate inflammatory and fibrotic factors, e.g., TGFβ and Notch signaling. Further factors include bone morphogenetic proteins (BMPs), thrombopoietin (TPO), platelet-derived growth factor (PDGF), and CCL3 (MIP1a) [86,87]. 

But what is known about OBC inflammatory function after HCT? One of the common long-term problems after allo-HCT is bone loss (reviewed in [88]), and one of the mechanisms includes the activation of OBCs by increased IL-6, IL-7, IFNg, and TNFa serum levels, predicting the amount of bone resorption during the first 12 months after allo-HCT [89,90]. In 2010, Shono et al. analyzed various cell populations affected by graft-versus-host disease (GVHD) and associated hematopoietic failure after HCT in different mouse models, in which GVHD was induced by targeted major histocompatibility complex (MHC)–mismatched transplantation [91]. They demonstrated that OBCs and not only HSPCs were the target of donor alloreactive CD4+ T cells. No simultaneous loss of ECs was observed. Given these deleterious interactions of OBCs and CD4+ T cells, anti-CD4 mAb therapies may be interesting to prevent bone loss after allo-HCT in MPN patients (illustrated in Figure 1B).

Contrary to MSCs and OBCs, ECs were found to harbor the *JAK2* c.1849G > T (V617F) mutation in 5 out of 22 MPN patients [92]. All 5 patients carrying the JAK2V617F mutation in ECs displayed thrombotic complications, increased phosphorylation levels of STAT3 and STAT5, and reduced therapy-free survival [92], suggesting a close link between JAK2V617F-positive ECs and thrombosis. Similarly, ECs of the spleen and splenic vein were described to be *JAK2*-mutated [93], and Edelmann et al. demonstrated in a JAK2V617F mouse model that β1 and β2 integrins of JAK2V617F-positive granulocytes revealed higher affinity than their normal counterparts to vascular cell adhesion molecule 1 (VCAM1) and intercellular adhesion molecule 1 (ICAM1), promoting venous thrombosis [94]. Therefore, thrombotic risk is affected by the malignant hematopoietic clone as well as aberrant ECs, showing increased adhesive properties and pro-inflammatory expression profiles (*CX3CL1*, *CXCL2*, *CXCR4*, *ALCAM*, *BST2*, and *SELP*), as also supported by another study [95]. Interestingly, when JAK2V617F expression was targeted exclusively to ECs in Tie2-Cre/FF1 mice, EC-mediated thrombo-hemorrhagic events were described [96]. Thus, depending on the specific microenvironment in MPNs, JAK2V617F + ECs may support anti-thrombotic or pro-thrombotic effects in MPN [97]. Nevertheless, just recently, Guy et al. were not able to detect the *JAK2*V617F mutation in PB-derived ECs of two MPN patient cohorts (DNA of 12/31 patients was analyzed), potentially demonstrating a lack of reproducibility of endothelial colony-forming cell (ECFC) cultivation techniques [98]. 

ECs after HCT remain for the majority of recipient origin, but a fraction of them will be of donor origin after a few months [99]. This finding confirms that a minority of transplanted HCs may retain endothelial differentiation potential, as is known for the hemangioblast during development. Sinusoidal obstruction syndrome (SOS), also called veno-occlusive disease (VOD), is induced by endothelial damage caused by the conditioning procedure and the release of exotoxins and cytokines in early transplant phase [80]. Patients with MF have a higher risk of SOS after HCT compared to those with other diseases [100]. It is unknown if the clonal origin of ECs in MF or increased levels of inflammatory factors in MF patients after HCT are responsible for this clinical observation.

Despite the progress in HCT for patients with high- or intermediate-risk MF, the graft failure rate remains high compared to other hematological diseases [101,102]. A higher rate of graft failure is also observed in patients receiving HCT for advanced myelodysplastic syndrome accompanied by fibrosis of the BM [103,104]. Furthermore, after HCT, donor-derived megakaryocytes often present dysplastic features such as microforms [105]. These clinical observations suggest that stromal dysfunction in MF is not always completely reversible when hematopoiesis is replaced by normal HSCs. Already in 2010, it was shown that BM-MSCs remain of recipient origin after HCT and do not harbor the JAK2V617F mutation in patients with MF [68]. However, MSCs have been described to carry other genetic and epigenetic aberrations, potentially induced by the long lasting interaction with the malignant hematopoietic clone [106,107]. Hence, the elimination of the malignant clone is not always sufficient for complete reversal of fibrosis of the BM and the inflammatory microenvironment is only partly modulated [108]. 

Reversal in cytokine and matrix modulation factors followed by regression of MF is expected after HCT, which is associated with better overall survival [74]. The exact role of most inflammation factors and mechanisms in remodeling ECM and fibrosis after HCT remains largely unknown, due to a lack of consecutive studies before and after HCT.

Hussein et al. studied the profile of cytokine and matrix modulation factors before and after HCT [109]. They focused particularly on fibrosis-inducing cytokines. At the RNA expression level, thrombospondin-1 (THBS1) as well as TIMP metallopeptidase inhibitor 1 (TIMP1) and 2 were elevated before HCT. In MF patients who had achieved remission, a decrease of TIMP1 and Platelet Derived Growth Factor Subunit A (PDGFA) was observed correlating with BM remodeling. However, other factors with a role in BM remodeling, such as TGFB2, SMADs, and BMPs, remained high, indicating ongoing remodeling. A biomarker for persistence of BM fibrosis was not identified. Furthermore, Hussein et al. concluded that resolution of fibrosis occurs mainly due to increased degradation rather than decreased production of extracellular matrix.

## 5. Therapeutic Approaches to Reduce Inflammation and to Prevent HCT Graft Failure

### 5.1. JAK Inhibitors in HCT 

The understanding of the development of MPN, mainly by unregulated activation of the JAK/STAT pathway and, hence, introduction of JAK inhibitors in the treatment of MPN, has fundamentally changed the management of MF patients before, during, and after HCT. Beyond the inhibition of the most abundant mutation, JAK2V617F, JAK inhibitors target the signal transduction of a broad range of cytokine receptors, which are signaling via associated JAKs after ligand binding, as described above [110]. Before the era of JAK inhibitors, no targeted treatment to reduce the cytokine-induced inflammation for all patients affected by MPN or also before HCT was available. Most patients had received hydroxyurea (HU) or no treatment during a long MF phase or before the start of conditioning in the context of HCT [111,112]. Some patients were splenectomized or received spleen irradiation to reduce the burden of splenomegaly, but morbidity and mortality of splenectomy were high [113]. Importantly, no clear advantage in the clinical outcome of HCT was found for splenectomized vs. non-splenectomized MF patients [114,115]. 

The JAK1/JAK2 inhibitor ruxolitinib (RUX) is the first JAK inhibitor approved by the FDA for the treatment of MF. In the COMFORT-I study, RUX vs. placebo was shown to significantly reduce spleen volume and constitutional symptoms in non-transplanted patients [116]. In the COMFORT-II study, RUX in comparison to best available therapy proved to be beneficial in reducing spleen volume and disease-related symptoms, and this was accompanied by an improved quality of life [117]. These effects are strongly related to changes in inflammatory cytokine-induced signaling (e.g., CRP, TNFa, MIP-1b, IL-1Rra, and IL-6), but a relevant reduction of mutant *JAK2* allele burden is rare, highlighting RUX as an immunomodulatory drug. Nevertheless, reduction in the mutant allele burden was observed in a fraction of patients [118], and BM fibrosis was reduced in a significant proportion of patients in long-term treatments with RUX [119]. In conclusion, the RUX effect was attributed to JAK1/JAK2 inhibition and not only JAK2 and, therefore, inflammatory stimuli are reduced. 

In the past years, RUX has also become standard treatment for the pre-HCT phase of two or three months before HCT [120] (Figure 2). The beneficial effect of RUX before HCT has been described by different study groups [121] and may be explained by the fact that patients enter the transplant phase under reduced inflammatory conditions and in an improved general status as well as with a reduced spleen size. All these factors may contribute to a better outcome of HCT in patients with MF. However, whether HCT outcome is truly improved by pre-HCT RUX is still being investigated in clinical trials. 

These outlined properties of RUX may also explain its efficacy post-HCT (Figure 2). RUX has an immunosuppressive effect via decrease of CD3-positive lymphocytes, activated T cells, as well as regulatory T cells [77,121]. Importantly, restoration of T cell numbers after HCT is not influenced by the treatment with RUX before transplantation [121]. In addition, RUX impairs NK cell number and function in MF patients [122].

Nevertheless, RUX needs to be withdrawn before the start of conditioning and during aplasia after HCT because of its hematotoxicity. Thus, RUX is usually stopped one day before the start of conditioning regimen. A rapid increase of JAK/STAT-dependent cytokines is observed, consequently, after the withdrawal of RUX. Tefferi et al. reported the “ruxolitinib withdrawal syndrome” in some patients after sudden RUX discontinuation, characterized by a loss of clinical benefit and induction of severe side effects [79]. Such patients suffered from a massive cytokine production with acute relapse of disease symptoms, accelerated splenomegaly, cytopenias, and occasional hemodynamic instability or death. Therefore, withdrawal of RUX before start of HCT conditioning must be cautiously planned. The chemotherapy included in most conditioning regimens mitigates the “ruxolitinib withdrawal syndrome” by destruction of the MPN clone and immune cells responsible for maintaining the inflammatory state. 

In a case report, Shiratori et al. observed a rapid and sustained increase in IL-6 and sIL-2R in two patients after stopping RUX despite the conditioning regimen with fludarabine and busulfan [123]. Both cytokines had decreased after the initiation of RUX a few months earlier. Other cytokines, such as vascular endothelial growth factor (VEGF), only showed a slight and temporary increase after RUX withdrawal. Nevertheless, no clinical symptoms of progression or increase in spleen size was observed after HCT, demonstrating the importance of an optimized RUX tapering strategy and immediate start of conditioning. 

In an attempt to establish recommendations on pre-transplant management, patient and donor selection as well as conditioning regimen, Kröger et al. summarized the available data by the European LeukemiaNet and European Blood and Marrow Transplantation Group [124]. Altogether, professional experience is mandatory to evaluate the decision for an HCT and optimize pre-transplant management, especially titration of RUX to the maximal tolerated dose and careful weaning until the day before conditioning. 

Despite the hematotoxicity of RUX, in one study, the drug was continued during conditioning and during the early peri-HCT phase [109]. The main reason to maintain the JAK inhibitor was a decrease in GVHD mediated by suppression of cytokine production (e.g., TNFa and IL-12p70) and T-cell expansion demonstrated in preclinical models [125,126]. The use of RUX in MF patients after HCT suffering from GVHD is of particular interest, as it is effective against the disease phenotype and the GVHD. In patients receiving RUX for several months before HCT, the medication was administered contemporaneous to the start of the conditioning, maintained until engraftment, and finally stopped around day 28 after a short tapering. Anti-thymocyte globulin (ATG) therapy was part of the conditioning regimen, and no increase of IL-8, IL-10, IL-6, TNFR2, and CD30/ TNFRSF8, but a significant decrease of IFNa and IFNb were observed in five analyzed patients compared to a historical patient cohort [109]. Neutrophil engraftment was observed at day 12 (d12) after HCT, which was sooner than expected compared to historical controls. The incidence of acute GVHD grade II-IV was only 8%. This study underlines the beneficial effect of RUX on the BM environment in patients with MF, with engraftment being improved despite the well-described hematotoxic effect of the drug. However, a high rate of cytomegalievirus (CMV) reactivation in this cohort also demonstrates the immunosuppressive properties of RUX.

The efficacy of RUX on GVHD was first described in 2015 [127], and a randomized trial has recently proven superiority of RUX compared to other treatment options for steroid refractory acute GVHD [81]. Efficacy of RUX in GVHD is attributed to the immunosuppressive properties of the drug and the inhibition of a wide number of cytokines involved in GVHD. However, the application of RUX in MF cannot be compared to the use of BCR-ABL-specific tyrosine kinase inhibitors (TKIs), such as imatinib, in patients transplanted for CML or Ph+ acute lymphoblastic leukemia (ALL) [128,129]. The efficacy of TKIs in CML and Ph+ ALL is attributed to a direct inhibition of the aberrant fusion protein, and a molecular response to the treatment is generally observed, whereas RUX is only acting on enhanced signal transduction without a significant decrease of the malignant clone [116], at least in most patients. Therefore, careful evaluation of patients with GVHD-associated inflammation is necessary before initiation of RUX treatment, as the immunosuppressive effect of RUX may also have severe side effects and favor relapse of the malignant disease and, eventually, clonal evolution [130]. Significant changes in several inflammation factors during and after HCT are illustrated in Figure 1 and Figure 2. 

### 5.2. Targeting ROS and Iron Overload

ROS play important roles in the progression of inflammation in MPN. As described above, NAC is efficient in MPN mouse models, and prophylactic administration of NAC promoted hematopoietic reconstitution even in a group of patients suffering from advanced myeloid neoplasms such as AML undergoing haploidentical HCT. In a fraction of these patients, a poor graft function and prolonged isolated thrombocytopenia were observed and defined as insufficient hematopoietic reconstitution post-HCT [131]. Interestingly, double-positive CD34+/CD309+ BM ECs were lower pre-HCT in patients with poor graft function and thrombocytopenia, and reconstitution of functional BM ECs was impaired post-HCT. These findings were positively correlated with increased ROS levels. Therefore, prophylactic NAC intervention was beneficial not only for the suppression of the malignant hematopoietic clone but also for the function of hematopoiesis-supporting BM microenvironmental cells, including ECs. Patients who develop GVHD after HCT often suffer from poor graft function. A higher concentration of ROS has been measured in endothelial progenitor cells (EPCs) in patients with GVHD by intracellular fluorescence detection after incubation with 2′,7′-dichlorofluorescence diacetate [132].

In another study using murine transplant models, donor-HSPC engraftment was ablated due to BM-localized inflammation mainly triggered by TNFa [133], a cytokine which is upregulated in MPN patients and MPN mouse models as described above [38,134,135]. Ishida and colleagues showed a significant increase of ROS in HSPCs upon TNFa treatment and reduced engraftment in primary and secondary transplants [133]. Pre-incubation with NAC reduced TNFa-triggered ROS production and normalized engraftment efficiency, again demonstrating the beneficial activity of this antioxidant. TNFa is a strong activator of the transcription factor NFkB and the serine/threonin kinase AKT, and both factors have been described to stimulate ROS production and inflammation [59,65,136]. It is conceivable that the use of dimethyl fumarate (DMF), an activator of nuclear factor erythroid 2-related factor 2 (Nrf2), may be beneficial in this setting as well. DMF is a ROS scavenger with anti-inflammatory as well as immunomodulatory properties. In a study by Han et al., DMF showed promising results not only in reducing acute GVHD but also in maintaining graft-versus-leukemia (GVL) effects [137]. However, DMF could be a two-edged sword, as the activation of redox-sensitive transcription factors, such as Nrf2, Bach1, and HIF1, is also described in MPN and AML [63,138]. 

ROS production is also connected to iron overload, which is a common problem in MPN patients due to the high frequency of transfusion-dependence for packed red blood cells (RBCs). Excessive iron leads to increased free iron, able to catalyze the conversion of ROS intermediates, such as superoxide anion (O_2_^−^) and hydrogen peroxide (H_2_O_2_) to highly toxic free radicals [139]. Pardanani and colleagues reported on increased levels of plasma hepcidin and serum ferritin in PMF patients, and both factors correlate with inferior survival independent of DIPSS-plus or increased inflammatory cytokine levels [140]. At least two studies on the impact of iron overload on HSC engraftment in mice were performed, one describing the impact of excessive iron on the BM niche and one on the HSCs. Okabe et al. generated iron-overloaded mice by the application of iron dextran and transplanted HSCs of normal mice [141]. They observed a significant delay in HSC engraftment and hematopoietic reconstitution and reduced expression of *Cxcl12*, *Vcam-1*, *Scf*, and *Igf-1* in BM stromal cells, suggesting a direct detrimental effect of iron on the BM microenvironment. Chai et al. caused iron overload in HSCs before transplant into normal mice, and observed impaired function of HSC long-term and multi-lineage engraftment [142]. In both studies, oxidative stress and ROS overproduction were confirmed, and detoxification with NAC or deferasirox (DFX) normalized engraftment and pluripotency. In conclusion, iron overload not only ablates HSC function, but also antagonizes BM niche functionality. Accordingly, an Italian group first retrospectively analyzed MF patients with iron overload, transfusion dependency, and DFX therapy and afterwards performed a clinical study with a similar patient cohort and DFX therapy [143,144]. Collectively, DFX treatment was found to be effective in hematological improvement and erythroid response in these studies. However, toxicity may be of concern, since a high incidence of adverse events was found in a cohort of pediatric HCT recipients during DFX therapy [145]. 

In summary, reducing oxidative stress and ROS before HCT may be a promising therapeutic approach to reduce inflammation and BM fibrosis as well to improve engraftment. Nevertheless, the physiologic activity of ROS on stemness and pluripotency of HSCs must be taken into account.

## 6. Conclusions

Experimental and clinical data confirm that inflammatory factors are involved in the pathogenesis of MPNs and in their progression to MF and acute leukemia. In addition to the mutated malignant clone, non-clonal HCs and BM microenvironmental cells, such as ECs, MSCs, and fibroblasts, undergo profound changes, and this reprogramming of non-hematopoietic cell types is involved in BM fibrosis. Among others, mutated dysplastic megakaryocytes are crucial for inflammatory cytokine secretion and recruitment of specific (non-mutated) MSCs and their differentiation into myofibroblasts, which in turn are responsible for fibrotic tissue formation. Remodeled OBCs are known to favor inflammation and stimulate fibrosis, and ECs (mutated as well as non-mutated) may be involved in higher thrombotic risk. Targeting only one of these cell types is challenging and at the same time may be not sufficient to eradicate the disease. Therefore, HCT is, although of a certain risk and with limitations, of high value for current MF therapy. Upon HCT, HSPCs and immune effector cells are replaced, expression of inflammatory factors are partially normalized, and fibrotic tissue in the BM can regress. However, several inflammatory factors remain upregulated after HCT, and no consistent biomarker has so far been identified to be associated with relapse or maintenance of fibrosis, demonstrating that some abnormalities induced by the previous clonal disorder may be irreversible. 

A higher graft failure rate and incomplete regeneration after HCT are observed after HCT in MF patients, and the inflammatory milieu is at least partially responsible. Clinical and also experimental data show that stromal adaptations to the abnormal and inflammatory microenvironment partially persist after HCT. However, HCT remains the only curative treatment for MPN, replacing dysfunctional malignant hematopoiesis by a new productive hematopoietic and immune system with encouraging long-term results in an increasing fraction of MF patients. Constant optimization of treatment regimens before and after HCT, targeting the survival of mutated HCs but also inflammation per se, with well-established inhibitors, such as RUX, but also compounds in trial, such as ROS scavengers, is still of crucial importance. 

## Figures and Tables

**Figure 1 cancers-12-02250-f001:**
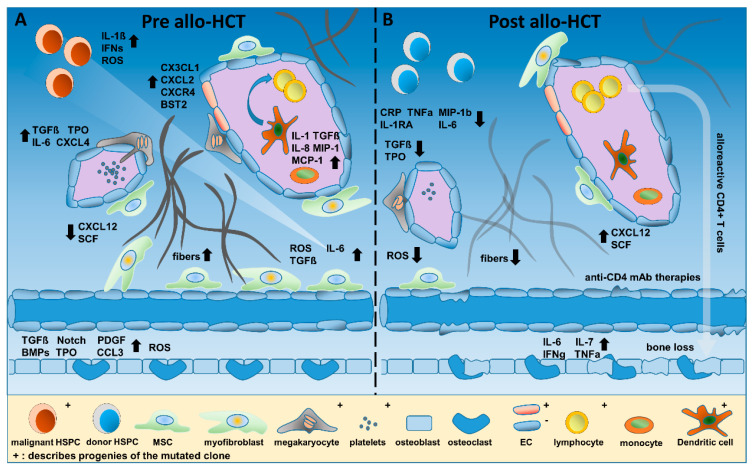
Overview of inflammatory cytokines and their producer cells before and after allogeneic hematopoietic (stem) cell transplantation (allo-HCT). (**A**) The pre allo-HCT microenvironment in myelofibrosis (MF) represents a state of inflammation in the hematopoietic niche of the bone marrow (BM). The malignant hematopoietic stem cells (HSCs), harboring the JAK2V617F, CALR, or MPL oncoprotein, secrete cytokines involved in inflammation, and they differentiate into aberrant hematopoietic progenitors such as megakaryocytes (and platelets), monocytes, dendritic cells, and granulocytes. Together, these cells induce BM stromal cells to secrete fibrogenic cytokines (e.g., TGFβ, IL-6, IL-1β, IL-8, and MIP-1). In addition, endothelial cells (ECs) may harbor driver oncogenes (confirmed for the JAK2V617F mutation), and these malignant ECs were demonstrated to trigger thrombotic events. Mesenchymal stromal cells (MSCs) are recruited to the niche by megakaryocyte-derived CXCL4 and differentiate into myofibroblasts, accelerating fibrosis via reticulin and collagen deposition. This leads to loss of hematopoietic support by the MSCs and subsequent suppression of normal blood cell production. Several cell types generate reactive oxygen species (ROS) in MF, which stimulate the proliferation of the malignant clone, induce DNA double-strand breaks, and correlate with complications in hematopoietic cell transplantation (HCT). (**B**) After conditioning and HCT, the malignant HSCs are depleted, and donor HSCs differentiate and replace aberrant hematopoietic but also certain non-hematopoietic cell types. The result is a reduction of MPN-associated inflammatory cytokines, ROS, and fibrosis and an increase of essential stemness factors, such as CXCL12 and stem cell factor (SCF). Alloreactive CD4+ T cells are involved in the graft-versus-fibrosis effect, but also in bone loss. ECs, osteoblasts, and osteoclasts are harmed in the process of HCT, and data are currently lacking to inform us whether their inflammatory activity is fully reversed. Whether myofibroblasts and mutated ECs undergo apoptosis or senescence after HCT has not yet been elucidated.

**Figure 2 cancers-12-02250-f002:**
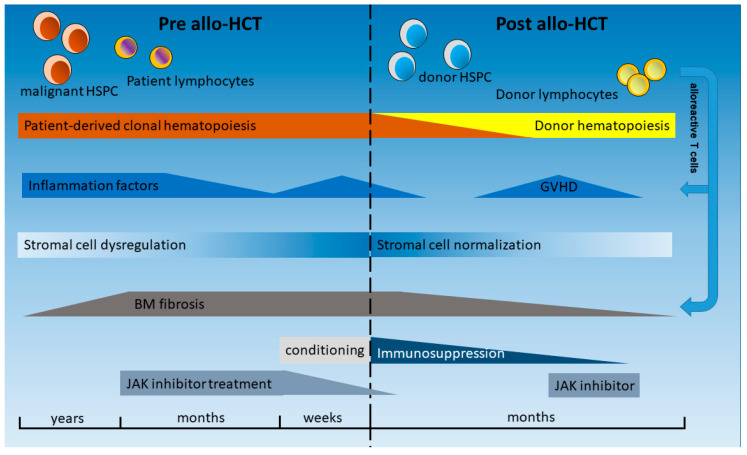
Order of events in the bone marrow (BM) before and after allogeneic hematopoietic stem cell transplantation (allo-HCT) in myelofibrosis. The malignant hematopoietic clone is considered as the origin of subsequent inflammatory alterations and modifications of stromal cells and the extracellular matrix [35]. These alterations finally lead to BM fibrosis [1]. Treatment with JAK inhibitors, such as ruxolitinib, can significantly decrease inflammation but, in most cases, not fibrosis [77,78]. The termination of ruxolitinib treatment and the conditioning regimen before HCT may induce a new increase in inflammation factors and ruxolitinib withdrawal syndrome [79] or sinusoidal obstructive syndrome [80]. After HCT, the elimination of the malignant clone and subsequent replacement by normal donor hematopoietic stem and immune effector cells typically lead to a persistent decrease in inflammation factors [74]. Over time, stromal cells often regain normal functionality, with normalization of extracellular matrix deposition and disappearance of fibrosis. Immunosuppressive medication will be tapered over time and stopped in the majority of patients. However, graft-versus-host disease (GVHD) can lead to an increase of inflammation factors, many of which are similar to the MPN-associated cytokines, albeit without an increase in fibrosis. Accordingly, JAK inhibitors can be used as immunosuppressive drugs in patients with refractory GVHD [81].

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
