# Peer review of "Role of Inflammatory Factors during Disease Pathogenesis and Stem Cell Transplantation in Myeloproliferative Neoplasms"

_cancers, 2020, doi:10.3390/cancers12082250_

Round 1

Reviewer 1 Report

This Review is aimed at reporting the impact of inflammation on clonal evolution/disease progression and describing the changes of the inflammatory microenvironment before and after hemopoietic stem cell transplantation in MPN patients. Even though many aspects emerge that indicate how inflammation contribute to the pathogenesis of MPN, the Review falls short in providing accurate and on target information. Moreover the conclusions of this review does not seem convincingly supported.

First of all, the Review describe a mixture of items without any logical sequence.

In addition, despite the title aim, few biological data/mechanisms have been reported about the characterization of the hemopoietic stem progenitor cells and crucial factors of the inflammatory bone marrow niche before and after hemopoietic stem cell transplantation in MPN patients. This might be due to the fact that the biological mechanisms are still largerly unknown. The majority of the data reported refer to the effects of hemopoietic stem cell transplantation on ther type of hematological disorders.

Major point is represented by Figure 1, where inflammatory mechanisms of the hemopoietic compartment and of the bone marrow niche before and after hemopoietic stem cell transplantation in MPN are shown without any supporting explanation in the text and without specific references.

Author Response

Dear Ms Wang,

We thank you and the referees for the review of our manuscript with the current title “Changes in circulating inflammatory factors during disease pathogenesis and stem cell transplantation in MPN” (Manuscript ID: cancers-868589). Based on this review, we have performed extensive changes to our manuscript including a strengthening of the focus, more detailed descriptions of individual inflammatory factors, and a modification of the title to highlight the more concise scope of the manuscript (see also point-by-point reply below). We are convinced that the review process has significantly improved the quality of our manuscript and hope that the manuscript is now acceptable for publication in “Cancers”.

Below, you will find our responses to the reviewers’ comments.

Sincerely,

Nicolas Chatain, Steffen Koschmieder & Edgar Jost

Reviewer 1

We thank the reviewer for his comments on our manuscript. We intensively optimized the manuscript accordingly and we wish to respond to the comments of the reviewer:

First of all, the Review describe a mixture of items without any logical sequence.

We agree that the logical sequence was clearly suboptimal, and the topics too diverse. Thus, according to the remarks of the reviewer, we have now strengthened both the focus and the logical sequence and structure of the manuscript. Also, in keeping with this more condensed scope, we have adjusted the title and abstract to be more concise in the topics of our review. In detail, we have reduced the amount of subchapters of the manuscript and have emphasized the focus on the role of inflammation in the pathogenesis of MPNs as well as after allogeneic hematopoietic cell transplantation (HCT). Thereby, we have taken great care in relating the content to these two major topics, including the use of JAK inhibitors and the role of ROS and antioxidants, and we have removed sections that were not at the center of our scope, such as clonal evolution, the role of IMIDs and disease progression.

In addition, despite the title aim, few biological data/mechanisms have been reported about the characterization of the hemopoietic stem progenitor cells and crucial factors of the inflammatory bone marrow niche before and after hemopoietic stem cell transplantation in MPN patients. This might be due to the fact that the biological mechanisms are still largerly unknown. The majority of the data reported refer to the effects of hemopoietic stem cell transplantation on their type of hematological disorders.

Indeed, the exact role of hematopoietic as well as stromal cells and the underlying mechanisms for the upregulation of inflammatory factors in MPN patients has not always been defined, including the post-transplant setting. On the other hand, data exist from both mouse models and humans on several specific inflammatory cytokines and cell subtypes, including fibrogenic cells. Therefore, we have now assembled more such data from mouse models and from patients, including data that have not exclusively been obtained in MPN models and patients but also in other closely related myeloid neoplasms. However, also in these cases, the important translation to MPN has been incorporated into the manuscript.

Specifically, we have expanded on the following topics:

  • In chapter 2 of the new manuscript, we now provide more specific information on the pathogenesis of inflammation in MPN and the cytokines and cell types involved.
  • The chapter on the role of the spleen after HCT was removed from the manuscript because detailed information is limited. Similarly, we have deleted the chapter on clonal hematopoiesis and IMIDs due to the lack of combined current information on inflammation and HCT.
  • During the process, the chapters on the effects of HCT on MPN, inflammation and matrix remodeling have been extended and restructured, as can be seen in the new chapters 3 and 4.

Major point is represented by Figure 1, where inflammatory mechanisms of the hemopoietic compartment and of the bone marrow niche before and after hemopoietic stem cell transplantation in MPN are shown without any supporting explanation in the text and without specific references.

The former Figure 1 is now Figure 2, and this figure has now been mentioned at several locations in the manuscript, and a timeline has been added to this figure.

Moreover the conclusions of this review does not seem convincingly supported.

We clearly agree with the reviewer, and, accordingly, have now re-written the conclusions to be more specific and focused on the inflammatory factors and their origin in MPN as well as how they are changed upon HCT.

Reviewer 2 Report

The manuscript "Circulating inflammatory factors during clonal evolution, disease progression and stem cell transplantation in MPN", written by Chatain N, Koschieder S and Jost E, describes changes in cytokines and other inflammatory factors during development of the myeloproliferative neoplasm, as well as their modulation after stem cell transplantation and targeted therapy. In the first introductory part (Background), basic types of MPN are described. Further on, the manuscript describes the consequences of the MPN linked to JAK mutations or those leading to similar effects. Several topics are presented in details: effects of stem cell transplantation on myelofibrosis, changes in inflammatory cytokines after transplantation, persistence of the fibrosis, changes in the stem cell niche after transplantation, effects of oxygen species, effects of the JAK inhibitors and the role of the spleen in MPN etc.

The manuscript is comprehensive, with numerous examples and references. However, to my opinion, it could be more systematized and better organized. It seems, when you read it, that the main topic are the effects of the different types of the therapy (such as transplantation and JAK inhibition) on the changes in the inflammatory factors in MPN and on the  myelofibrosis. The Abstract summarizes the content and mentions inflammatory processes in only few sentences, therefore does not correspond to the title of the manuscript. The facts considering regression of myelofibrosis (79-80 line) are repeated in the Background and in the next paragraph. The paragraph "Reactive oxygen species and iron" seems to me not to be directly related to the topic (it could be enclosed in the paragraph describing persistence and progression of the disease). Also, considering description of different chemotherapeutics and i. e. ROS inhibitors, there are details on the experiments on the mice, but it could be also mentioned whether they were in clinical trials.

I would suggest to write first about inflammatory factors in the disease (as in paragraph 3), role of  the spleen, niche, and cofactors influencing disease progression, and then about changes after different types of therapy, such as HCT and effects of the JAK inhibition, with the accent on the circulating inflammatory factors' presence.

Minor mistakes

Abstract: line 22: mouse models; iPS cells

line 24: However

line 135: were demonstrated

line139: clone may acquire

line 184: better sentence construction

line 234 there were not these mutations

lines 380-381: better sentence construction

lines 287, 432, 453, 487, 719: changes in font and spacing

Author Response

Dear Ms Wang,

We thank you and the referees for the review of our manuscript with the current title “Changes in circulating inflammatory factors during disease pathogenesis and stem cell transplantation in MPN” (Manuscript ID: cancers-868589). Based on this review, we have performed extensive changes to our manuscript including a strengthening of the focus, more detailed descriptions of individual inflammatory factors, and a modification of the title to highlight the more concise scope of the manuscript (see also point-by-point reply below). We are convinced that the review process has significantly improved the quality of our manuscript and hope that the manuscript is now acceptable for publication in “Cancers”.

Below, you will find our responses to the reviewers’ comments.

Sincerely,

Nicolas Chatain, Steffen Koschmieder & Edgar Jost

Reviewer 2

We thank the reviewer for his detailed comments. According to the suggestions of the reviewer, we have reorganized the manuscript.

The manuscript is comprehensive, with numerous examples and references. However, to my opinion, it could be more systematized and better organized. It seems, when you read it, that the main topic are the effects of the different types of the therapy (such as transplantation and JAK inhibition) on the changes in the inflammatory factors in MPN and on the  myelofibrosis.

We agree with the reviewer and have now emphasized the data available on those cell types known to play a role in the increase of inflammatory factors in MPN and how these factors may influence disease pathogenesis in the first part of the manuscript. In the second part, we have now highlighted the changes in the niche after HCT with a focus on the available information of the alterations in circulating cytokines, which may be involved in fibrotic remodeling. In the following, current optimization of the treatment regimens during HCT with a particular emphasis on ruxolitinib and ROS scavengers (as a potential future treatment option) are discussed.

The Abstract summarizes the content and mentions inflammatory processes in only few sentences, therefore does not correspond to the title of the manuscript.

We have changed the title of the review to present a more concise review on the changes of inflammatory factors during MPN pathogenesis and their potential reversal or irreversibility upon HCT. We have also adjusted the abstract, as suggested by the reviewer, emphasizing the focus of this manuscript on inflammation in MPN and how inflammation can influence the success of HCT. As a result, treatment options remain important in the manuscript; however, they are put into closer context with HCT and the manipulation of the inflammatory microenvironment.

The facts considering regression of myelofibrosis (79-80 line) are repeated in the Background and in the next paragraph. 

We are sorry for this inattention and have now corrected the respective passage.

Further changes have been applied according to the reviewers’ comments:

  • To be more concise, we have removed some parts of the manuscript (as also outlined above in the comments to reviewer 1), such as the role of the spleen, clonal hematopoiesis (no direct connection to HCT) and IMIDs.

The paragraph "Reactive oxygen species and iron" seems to me not to be directly related to the topic (it could be enclosed in the paragraph describing persistence and progression of the disease). Also, considering description of different chemotherapeutics and i. e. ROS inhibitors, there are details on the experiments on the mice, but it could be also mentioned whether they were in clinical trials.

  • The paragraph on reactive oxygen species (ROS) has now been divided, and parts are now included in chapter 2 on “Inflammatory cytokines and their sources in MPN” and in chapter 4 “Reduction of inflammatory stimuli for the prevention of HCT graft failure”. In addition, we have aimed at highlighting the available clinical and pre-clinical data on ROS scavengers. ROS have been demonstrated to play a key role in inflammation and most likely also in disease progression in MPN. This is the reason why we have included this chapter in the review and we explained the therapeutic possibilities related to HCT and engraftment in MPN patients.
  • Minor changes: all minor mistakes mentioned by the reviewer have now been corrected in the revised manuscript.

Round 2

Reviewer 1 Report

The revised manuscript is now well organized and focused on the aims of the Review.

However, the title needs to be changed. It doesn' t seem totally convincing and understandable. What does it mean " Changes in circulating inflammatory factors during disease pathogenesis......." ?

Author Response

We thank the reviewer for his comment.

We changed the title of the manuscript to "Role of inflammatory factors during disease pathogenesis and stem cell transplantation in myeloproliferative neoplasms"

Reviewer 2 Report

The authors accepted the suggestions and comments given in the comments They reorganized and rewrote the manuscript and I would suggest its acceptance for publication.

Minor: I would suggest to omit the drug concentrations in the ruxolitinib therapy (line 385).

Author Response

We thank the reviewer for his comment.

According to the suggestions of the reviewer, we have skipped the dosage of ruxolitinib in line 385.